# Atrial Fibrillation Related Coronary Embolism: Diagnosis in the Focus

**DOI:** 10.3390/jpm13050780

**Published:** 2023-04-30

**Authors:** László Balogh, Péter Óvári, Christopher Uwaafo Ugbodaga, Zoltán Csanádi

**Affiliations:** Department of Cardiology and Cardiac Surgery, Faculty of Medicine, University of Debrecen, 4032 Debrecen, Hungary

**Keywords:** coronary embolism, atrial fibrillation, personalized thrombus aspiration

## Abstract

Atrial fibrillation (AF) is the most common arrhythmia in myocardial infarction (MI). AF can be caused by ischemia, and MI can be caused by AF. Additionally, 4–5% of MI cases are related to coronary embolism (CE), and one-third of cases are attributed to AF. Our aim was to investigate the prevalence of AF-related CE cases among 3 consecutive years of STEMI cases. We also aimed to reveal the diagnostic accuracy of the Shibata criteria scoring system and the role of thrombus aspiration. Among 1181 STEMI patients, 157 had AF (13.2%). By using the Shibata’s diagnostic criteria, 10 cases were classified as ‘definitive’ and 31 as ‘probable’ CE. After re-evaluation, a further five cases were classified as ‘definitive’. Further analysis of the 15 CE cases revealed that CE was more prevalent in patients with previously known (n = 10) compared to those with new-onset (n = 5) AF (16.7% vs. 5.1%, *p =* 0.024). A PubMed search was performed, and 40 AF-related cases were found where the Shibata’s criteria could be applied. Further, 31 cases could be classified as ‘definitive’, 4 as ‘probable’ and, in 5 cases, the embolic origin could be excluded. In 40% of reported cases and in 47% of our cases, thrombus aspiration helped in diagnosis.

## 1. Introduction

Atrial fibrillation (AF) is the most common arrhythmia in adults (1–2%), with an incidence of 9% in the elderly population (>80 years) [1], and 10–15% of AF patients undergo PCI for coronary artery disease because they share many risk factors. The incidence of AF in acute myocardial infarction (AMI) ranges from 2 to 23%, and AF might be associated with an increased risk of AMI [2]. It has been estimated that 6–12 million people will suffer from this condition in the US by 2050 and 17.9 million people in Europe by 2060 [3,4]. AF frequently complicates acute coronary syndromes, especially acute ST-segment elevation myocardial infarction (STEMI). Ischemic-damage-induced heart failure is the most important predictor of the development of AF related to AMI [5]. 

Over the last few decades, the treatment for AMI has changed. During the first decade of this millennium, reperfusion therapy turned from fibrinolysis to primary percutaneous coronary intervention, with a concomitant change in medical therapy. Thus, the reviewed incidence of AMI-related AF has also changed. Prior to the thrombolytic era, the incidence of AF was reported as 18% of AMI cases, which decreased to 9.8%, 10.4% and 6.8% reported in a nationwide survey, in the GUSTO I and GUSTO III trials, respectively, mostly due to the more widespread use of systemic thrombolysis [6,7,8,9]. Other studies conducted in the PCI era reported the incidence of acute atrial fibrillation between 4.6 and 9.4% [10]. A retrospective analysis of the AMI and AF relationship, conducted between 2011 and 2015, revealed an increased rate of AMI in AF patients compared to those who were in sinus rhythm (12.0% vs. 6.0%) [11]. AMI-associated AF is a previously known AF (PAF) and can be new-onset AF (NOAF) related to AMI. 

AF can be caused by acute myocardial infarction (AMI) and, conversely, AMI can be caused by AF due to coronary embolism (CE). CE is an important non-atherosclerotic cause of AMI. Nevertheless, both AF and AMI can occur independently. The exact prevalence of CE remains unknown because it is difficult to diagnose in the acute phase of coronary occlusion. In a study published in 2015, AF was reported as the most frequent cause of CE, with a prevalence of 2.9% in patients with AMI [12]. In this study, Shibata T. et al. proposed a diagnostic algorithm for CE, which contains major and minor criteria mainly concentrating on angiographic findings, risk factors and source of embolism. Popovic B. et al. studied 1232 consecutive patients who presented with de novo ST-segment elevation myocardial infarction and confirmed 53 patients (4.3%) with coronary embolism by using Shibata’s diagnostic system [13]. The prognosis and mortality of acute myocardial infarction are poor in the presence of atrial fibrillation [14]. The prognosis of embolic myocardial infarction is poorer compared to plaque-rupture-related thrombotic occlusion, due to the lack of preconditioning and collateral formation in CE. In a recently published systematic case review, Lacey M.J. et al. reported 12.9% mortality in CE [15]. 

Therefore, the present study was designed to evaluate the prevalence of AF and to identify AF-related CE cases in consecutive STEMI cases during a 3-year period. We aimed to investigate the bidirectional relationship between AF and AMI. We aimed to assess the diagnostic accuracy of the Shibata criteria system in our cohort. We wanted to investigate the sensitivity, specificity and diagnostic accuracy of the Shibata criteria system to propose whether refinement is required. Many AF-related AMI cases (mainly case reports) have been listed so far in the literature. Assuming that these were definite AF-related embolic cases, it seemed obvious to test those with Shibata’s criteria to identify their diagnostic accuracy. Therefore, AF-related CE cases published in the last 15 years were investigated by using Shibata’s criteria. 

Thrombus aspiration was downgraded in the recent guidelines [16], but it might help to identify embolic cases. In a study, routine aspiration thrombectomy showed better recognition of embolic AMI, allowing for the avoidance of stenting [17]. We also aimed to assign a subpopulation in AMI where the personalized use of thrombus aspiration might help to secure the diagnosis, even if thrombus aspiration has no mortality benefit and is routinely not recommended according to recent guidelines.

## 2. Patients and Methods

In this study, 1181 consecutive patients with ST-elevation myocardial infarction (STEMI) treated in our institute between 1st of January 2014 and 31st of December 2016 were analysed retrospectively to identify patients with AF (n = 157). Patient files and coronary angiograms were evaluated by two independent, experienced interventional cardiologists. The cardiologists were asked to categorize the cases into groups of low or high probability of coronary embolism. Patients were divided into previously known (paroxysmal or permanent, PAF) and acute coronary syndrome-related atrial fibrillation (new onset, NOAF) groups. CHA_2_DS_2_-VASc score was calculated in patients with and without AF to identify the risk for systemic embolization. Thrombus aspiration activity in different subgroups was also recorded. 

The PubMed database was searched using the keywords ‘coronary, embolism, myocardial, infarction’ to find relevant case reports for coronary embolism to test the diagnostic accuracy of Shibata criteria. The AF-related AMI cases were used as a gold standard to test the diagnostic accuracy of Shibata criteria. Thus, 3381 articles were found between 2005 and 2019. After a manual search, 259 relevant case reports (307 cases) were found; among them, 44 cases were AF-related. There was enough information in 40 case reports for further testing of the criteria system (see Appendix A).

The following major and minor criteria proposed by Shibata were used to identify potential CE and applied for our patient cohort as well as for patients presented in case reports on PubMed:Major criteria:Angiographic evidence of coronary artery embolism and thrombosis without atherosclerotic components.Concomitant coronary artery embolization at multiple sites (multiple vessels within 1 coronary artery territory or multiple vessels in the coronary tree).Concomitant systemic embolization without left ventricular thrombus attributable to acute myocardial infarction.Minor criteria:<25% stenosis on coronary angiography, except for the culprit lesion.Evidence of an embolic source based on transthoracic echocardiography, transoesophageal echocardiography, computed tomography or MRI.Presence of embolic risk factors: atrial fibrillation, cardiomyopathy, rheumatic valve disease, prosthetic heart valve, patent foramen ovale, atrial septal defect, history of cardiac surgery, infective endocarditis or hypercoagulable state.

CE was classified as ‘definite’ or ‘probable’ according to the following criteria (one of the following): 

Definite CE:
Two or more major criteria.One major criterion plus ≥two minor criteria.Three minor criteria.

Probable CE:
One major criterion plus one minor criterion.Two minor criteria.

A diagnosis of CE should not be made in the presence of:
Pathological evidence of atherosclerotic thrombus.History of coronary revascularization.Coronary artery ectasia.Plaque disruption or erosion detected via intravascular ultrasound or optic coherence tomography in the proximal part of the culprit lesion.

The current proposed diagnostic criteria for CE include 3 major and 3 minor criteria. A weighted scoring of the criteria is used to differentiate between definite and probable CE in patients with AMI. 

## 3. Statistical Analysis

The distribution of parameters was examined using the Kolmogorov–Smirnov test. Variables showing normal distribution were expressed as mean ± SD, while parameters with non-normal distribution were described as medians. Between group differences were analyzed using Student *t*-test when normally distributed or by Mann–Whitney test in the case of non-normal distribution. Differences in category frequency were evaluated via χ^2^ test. A *p* value of less than 0.05 was considered to be statistically significant. The Statistical Package for the Social Sciences (SPSS 26, Chicago, IL, USA) was used for statistical analyses.

## 4. Results

Among 1181 STEMI patients, 157 had AF (13.2%). Further, 60 (5.1%) patients had previously known AF (PAF), and 97 (8.2%) could be categorized into AMI-related or new-onset AF (NOAF) groups (Figure 1). Shibata’s criteria were used to detect potential embolic myocardial infarction cases. 

Among all AF cases, an embolic origin of AMI was excluded in 116 patients, and 31 and 10 were categorized as ‘probable’ and ‘definite’, respectively. Among patients with PAF, embolic origin could be excluded in 38 cases; 15 cases were categorized as ‘probable’ and 7 cases as ‘definite’. Among NOAF patients, the diagnosis of CE could be rejected in 78 cases, 16 cases classified as ‘probable’ and 3 cases as ‘definite’ by using Shibata’s criteria (Figure 1). 

A detailed analysis of the patients’ files and coronary angiograms of patients who had AF (n = 157) was conducted by two experienced interventional cardiologists revealed that ‘definite’ cases using Shibata’s criteria could be categorized into high probability regarding CE in both the PAF and NOAF groups. Additionally, 4 of the ‘probable’ (2 in PAF and 2 in NOAF) cases among 31 (15 PAF and 16 NOAF) cases classified as ‘probable’ according to Shibata’s system had high clinical and angiographic CE probability. In the remaining 27 (13 PAF and 14 NOAF) cases, the diagnosis was rejected (low probability) due to visible plaque rupture at the culprit site (Figure 2). 

In one case in the ‘unlikely’ category, the diagnosis of CE could be confirmed (Figure 2). In this case, the patient previously had coronary artery bypass grafting. According to Shibata, the diagnosis of CE should not be made; however, the angiographic suspicion of CE was very high. 

Among 157 patients with AF, 15 had CE (9.6%). Thereafter we worked with these 15 CE cases (10 identified by Shibata + 5 cases found after re-evaluation). Further analysis of the AF subgroups revealed that CE was more prevalent among patients with PAF (n = 10) compared to those with NOAF (n = 5) (16.7% vs. 5.1%, *p =* 0.024). Thus, 7 out of 10 CE cases emerged from previously known permanent AF patients (n = 29) and the remaining 3 from previously known paroxysmal AF patients (n = 31) (24.1% vs. 9.7%, *p =* 0.175). It suggests that CE cases may occur more frequently in patients with PAF, even if the difference was not statistically significant (Figure 3). 

In all our patients with AF, the CHA2DS2-VASc score was calculated. The CHA2DS2-VASc score was significantly higher in NOAF compared to a patient who had no AF at all (2.7 vs. 2.1, *p =* 0.01). The CHA2DS2-VASc score was significantly higher in PAF compared to NOAF patients (3.7 vs. 2.7, *p <* 0.0001). The CHA2DS2-VASc score was higher in patients with permanent compared to paroxysmal AF (4.2 vs. 3.2, respectively, *p =* 0.83), even if the difference was not statistically significant (Figure 4).

In the whole cohort (n = 1181), thrombus aspiration was performed in 295 cases (24.9%) and in the AF cohort (n = 157), in 43 cases (27.4%). There was no significant difference in thrombus aspiration tendency among different AF subgroups (Figure 4). The effectiveness of anticoagulation was also investigated, but no significant difference was found between the CE and non-CE groups (data not shown).

The literature search for AF-related CE cases resulted in 40 relevant cases. By using Shibata’s criteria, 31 cases (77.5%) were classified as ‘definite’, 4 cases (10%) as ‘probable’ and, in 5 cases (12.5%), the diagnosis of coronary embolism had to be rejected (Figure 5). It was also noticed that in 23 out of the 40 case reports, thrombus aspiration was performed. In 16 cases, the diagnosis of CE was based on the aspiration results, and there was no angiographic sign of plaque disruption or stenosis at the site of occlusion following thrombus aspiration. In our cohort, the diagnosis of CE could be confirmed via aspiration in 7 patients out of 15 (46.6%).

## 5. Discussion

CE is frequent enough (4–5%) to be considered in all cases of AMI, especially if an embolic risk factor (AF), source (left-sided rheumatic valve disease, left atrial myxoma, mechanic heart valve, infective endocarditis, ventricular aneurysm, dilated cardiomyopathy, aortic fibroelastoma, sinus Valsalva aneurysm) or conduit (patent foramen ovale (PFO) atrial septal defect (ASD)) is present to cause or mediate embolism. In the case of multiplex coronary and/or concomitant systemic arterial occlusions (e.g., AMI + stroke, AMI + pulmonary embolism + PFO/ASD, etc.), an embolic origin must be suspected. Stroke is not the only consequence of systemic thromboembolism. In addition to cerebral arteries, the coronaries can also be affected by systemic embolization. A lower incidence of coronary embolism (CE) can be explained by a lower percentage of cardiac output moving through the coronaries compared to cerebral circulation and by the fact that coronary orifices are partially covered by opening aortic valve cusps, and the ejected blood moves mainly parallel to the aortic wall during systole. However, during diastole, after closure of the aortic valve, the flow can easily turn to the direction of the coronary arteries where the flow is predominantly diastolic.

The co-incidence of AMI and AF is very frequent. AF-related left atrial appendage thrombus is the most common source of systemic embolism. Therefore, it is important to investigate the causal relationship between AMI and AF. It is true for previously known and new-onset cases as well. In our cohort (n = 1181), we found AF in 13.2% (n = 157) of cases, broken down into 8.2% (n = 97) NOAF and 5.1% (n = 60) PAF. Shibata et al. proposed a diagnostic criteria scoring system to identify coronary embolic cases. By using these criteria in our study, the cases with clinically high embolic probability could be defined precisely (‘definite’), 7 CE cases in the PAF and 3 in the NOAF group. After re-evaluation of the 10 cases, we could confirm the diagnosis of CE. However, in most patients categorized into the ‘probable’ group (15 PAF and 16 NOAF), the diagnosis of CE could not be confirmed by detailed analysis of the patient’s files and coronary angiograms. The diagnosis of CE was highly suspected after re-evaluation in only 4 of 31 cases (2 PAF and 2 NOAF), and certain signs of plaque rupture were found in the remaining 27 cases (13 PAF and 14 NOAF). Obvious signs of plaque rupture at the site of coronary occlusion almost certainly exclude the possibility of CE. In such cases, there is no causal relationship between AF and AMI. It points out that coronary angiograms in ‘probable’ CE cases must be re-evaluated. We also tried to find the reason why so many plaque-rupture-related cases were categorized into the ‘probable’ CE group. It has been noticed that a majority of cases are misclassified as ‘probable’, because, according to Shibata, if someone has AF and no coronary stenosis (<25%) except the culprit site, the patient falls into the ‘probable’ category. ‘AF’ and ‘<25% coronary artery stenosis except the culprit vessel’ criteria can frequently be coincidental and without embolic background. In our opinion, the oversensitivity of Shibata’s system and the unexpectedly high number of ‘probable’ cases are mainly attributed to this. In addition, we could also identify a case with high clinical probability for CE, which was rejected by the Shibata system. Shibata’s criteria reject the diagnosis of coronary embolism in cases of previous revascularization, even though the co-incidence of coronary stenosis and AF-related embolism is very common. Our case with previous bypass surgery and left internal mammary artery occlusion treated with only thrombus aspiration also underlines the importance of individual case evaluation since it could have been missed by using the criteria system alone. In this case, after thrombus aspiration, the flow completely restored, and stent implantation was not required due to a lack of visible vessel wall abnormality at the site of the occlusion. Clinically, it is very difficult to differentiate between locally formed and embolized coronary thrombus; however, it is crucial due to the distinct therapeutic approach. The discrimination of embolic acute myocardial infarction from plaque rupture or erosion-caused thrombus is hard in acute clinical setting. The most complicated issue is the identification of an embolized thrombus, which was lodged at a site of coronary stenosis. However, it occurs more frequently than it is thought to. Intravascular imaging is the only tool which can differentiate between locally (on ruptured plaque) formed and embolized thrombus, but it is rarely used in acute clinical settings and daily routine. After re-evaluation of AF cases, it could finally be concluded that 9.6% (n = 15) of all AF patients (n = 157) had embolic myocardial infarction. Further analysis of subgroups revealed that CE was significantly more prevalent in PAF (n = 10) compared to NOAF (n = 5) groups, which underlines the importance of the search for an embolic origin of coronary occlusion in the PAF group, especially when AF is permanent (n = 7). In the NOAF group, the CE incidence was 5.1%, which is close to the general incidence of CE among all AMI cases (4–5%). Occurrence of NOAF is a marker for atrial volume and pressure overload associated with heart failure in acute coronary syndrome. Tachycardia and irregular ventricular activity during AF may cause further impairment in the coronary perfusion and left ventricular function. This can raise the suspicion that the majority of NOAF cases are caused by myocardial ischemia and infarction-related factors affecting myocardial contractility and the resulting left atrial stretch. A meta-analysis demonstrated that older age and increased heart rate levels on admission are related to a greater risk of NOAF in AMI. Patients with NOAF have severe left ventricular dysfunction, cardiovascular risk factor burden, pulmonary hypertension, valvular disorders, and left atrial enlargement compared with patients without NOAF [10,11]. Subgroup analysis of our AF cases revealed obvious differences in the incidence of CE in different AF groups. In the subgroup of PAF, the incidence of CE vas 16.7%, which was significantly higher compared to NOAF or compared to the general incidence of CE without AF. Numerically, there was more CE in the permanent compared to paroxysmal AF group (24.1% vs. 9.7%), but the difference was not statistically significant, probably due to the low number of cases.

Shibata’s criteria were also applied on previously published AF-related CE cases to evaluate their diagnostic accuracy. A few cases (10%) were categorized as ‘probable’, and a few cases were missed (12.5%), but most of them (77.5%) were diagnosed as ‘definite’ CE. A relatively high proportion of ‘probable’ and missed cases raises the question if refinement of the criteria system is required. 

The CHA_2_DS_2_-VASc score can give the embolic risk in AF. This score was also tested previously on patients who had no AF, and it was found that it can predict new-onset AF; however, it is not routinely used for prediction in clinical practice [18,19]. We found significantly lower CHA_2_DS_2_-VASc scores in patients who had no atrial fibrillation compared to the NOAF group (2.1 vs. 2.7). The CHA_2_DS_2_-VASc score detected in PAF was significantly higher compared to the NOAF group (3.7 vs. 2.7). A gradual and significant increase in the CHA_2_DS_2_-VASc score from no AF through NOAF to PAF groups indicates a higher embolic risk and a higher chance for embolic myocardial infarction. These results are congruent with CE case numbers detected in different cohorts. The lowest CHA_2_DS_2_-VASc score and CE incidence among AF cases was detected in the NOAF group, and a gradual increase in the number of CE cases can be observed with a proportionally gradual increase in the CHA_2_DS_2_-VASc score in paroxysmal and permanent AF cases (2.7 vs. 3.2 vs. 4.2). Between NOAF and permanent PAF, the difference was significant, between paroxysmal and permanent AF subgroups, the previously mentioned differences were not statistically significant, maybe due to the low patient numbers in different subgroups.

Thrombus aspiration was a considerable option at the time of patient enrolment into our study (2014–2016 (IIb class indication). Following the neutral results of the TASTE study, its routine use is no longer recommended (III class indication) because of the high complication rate and neutral cardiovascular outcome [20]. However, aspiration is not prohibited in the current guidelines [16]. We think that the thrombus aspiration can be useful, not due to improvements in clinical outcome of the patients, but it might help to ensure the diagnosis of CE in suspected cases. By analyzing the case reports, we could conclude that in 40% of the cases (16 of 40), thrombus aspiration was crucial to ensure the diagnosis of CE. In our whole cohort (n = 1181), thrombus aspiration was performed in 24.9% of the cases. In AF-related cases, the thrombus aspiration was the cornerstone of the diagnosis in 7 out of 15 patients (46.6%). In almost half of the cases, thrombus aspiration helped to confirm the diagnosis of CE. Similar data can be found in the literature, demonstrating that half of the CE cases are missed without thrombus aspiration [17]. The question is whether there is any subgroup of patients who can benefit from thrombus aspiration. However, the detection of embolism is difficult without thrombus aspiration. We think that all CE cases can benefit from thrombus aspiration, not necessarily from clinical outcome but from a diagnostic point of view and planning of long-term treatment. To date, there have been no clinical trial aiming to answer the question whether thrombus aspiration provides better clinical outcome in CE cases, but it is easy to accept why thrombus aspiration (sometimes without further balloon dilation or stent implantation) can be reasonable in this scenario. We would suggest that thrombus aspiration can be considered after wire passage in AF patients with AMI. In definite embolic cases, dual anti-platelet therapy cannot prevent further embolic events. Therefore, unnecessary stenting must be avoided. These cases of acute coronary syndrome require a more personalized approach regarding anti-thrombotic treatment. Oral anticoagulation is the only option and the most effective long-term treatment and secondary prevention strategy for these patients.

Limitations of the study: There is no routinely used gold-standard method for the diagnosis of CE, which we would be able to compare our results to. The retrospective nature of the investigation makes it difficult to confirm or to rule out coronary embolism in an equivocal case. No systematic follow-up of patients was performed if AF or other embolic events occurred. In patients categorized to no AF groups, one cannot exclude unrecognized paroxysmal AF, as previously published AF-related CE case misdiagnosis cannot be excluded. Numerous sources of embolism can be present in patients but these are sometimes hard to identify. In equivocal cases, transesophageal echocardiography (TEE) might help to find the source, such as left atrial appendage thrombus, left atrial myxoma, aortic fibroelastoma or a conduit, such as patent foramen ovale (PFO) or atrial septal defect (ASD). In this retrospective study, TEE is not an option. 

## 6. Conclusions

The co-incidence of MI and AF is common and an investigation of the causal relationship between each of them is reasonable in all cases. Long-term outcomes indicate that CE patients represent a high-risk subpopulation of patients with AMI, therefore, requiring close follow-up. Hence, a more precise diagnosis of coronary embolism would be needed. The main learning points of our study: 1. Incidence of CE was 9.6% in all AF patients, and incidence of CE was significantly higher in the PAF compared to the NOAF group. The CE incidence in the NOAF group was close to the incidence of CE among all STEMI cases. 2. Shibata’s criteria are sensitive enough to identify CE cases, but their specificity and diagnostic accuracy in the ‘probable’ category are lower than expected; therefore, refinement of the criteria system and and/or re-evaluation of suspected cases are required. 3. Routine use of thrombus aspiration is not recommended in STEMI, but a subgroup of patients (e.g., PAF) may benefit from thrombus aspiration. Aspiration helps in diagnosis and in the planning of long-term drug treatment. Without thrombus aspiration, the diagnosis of almost half of CE cases would be missed. Patients with AMI and coexisting AF are less likely to receive appropriate therapy and more likely to experience adverse outcomes than AMI patients in sinus rhythm. 

## Figures and Tables

**Figure 1 jpm-13-00780-f001:**
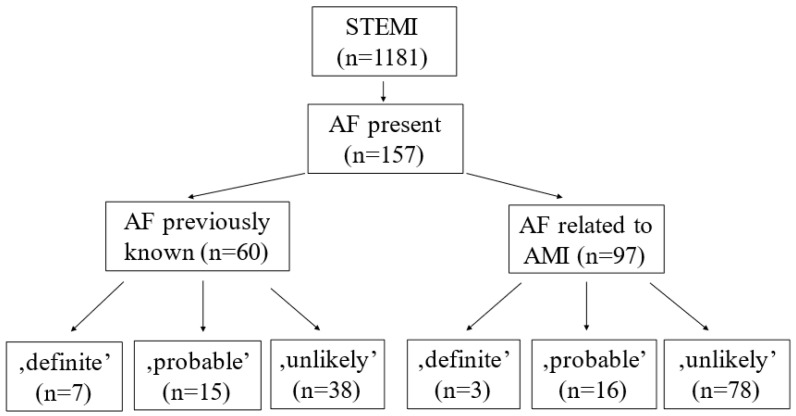
Probability of coronary embolism in patients with previously known and AMI-related atrial fibrillation.

**Figure 2 jpm-13-00780-f002:**
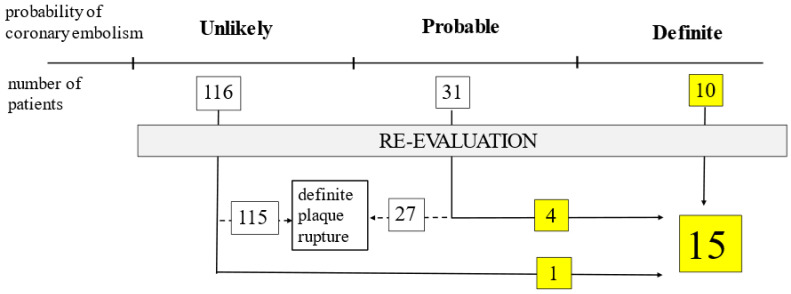
Results of re-evaluation of all atrial fibrillation cases (n = 157) regarding coronary embolism based on Shibata’s criteria.

**Figure 3 jpm-13-00780-f003:**
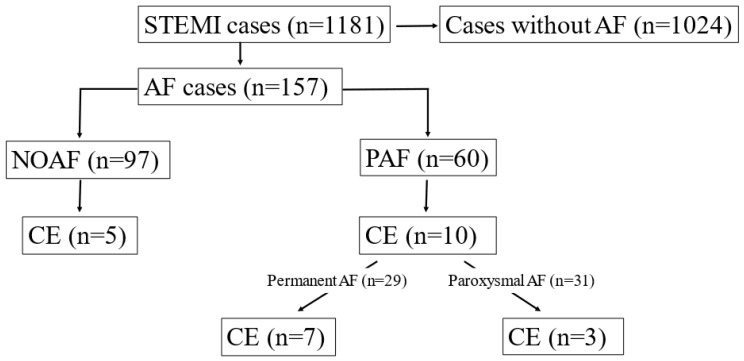
Prevalence of coronary embolism (CE) among different subgroups of patients with atrial fibrillation (AF).

**Figure 4 jpm-13-00780-f004:**
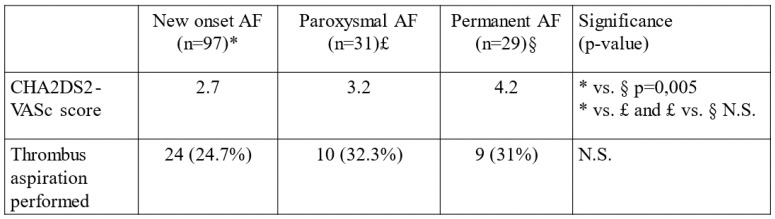
CHA2DS2-VASc score values and number of thrombus aspirations performed among NOAF, paroxysmal and permanent PAF subgroups. * = NOAF, £ = paroxysmal AF, § = permanent AF.

**Figure 5 jpm-13-00780-f005:**
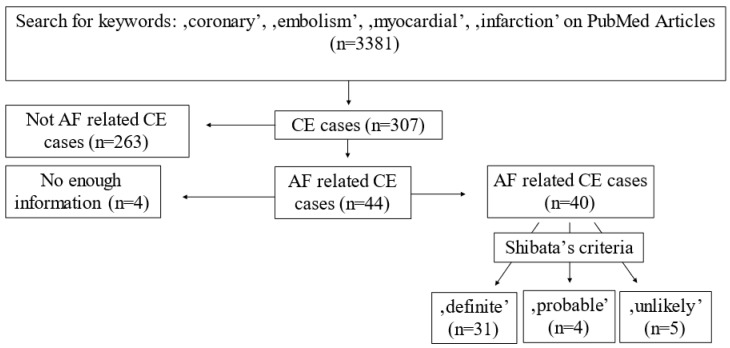
Literature search regarding atrial-fibrillation-related coronary embolism and application of the criteria proposed by Shibata.

## Data Availability

Not applicable.

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
