# Peer review of "Atrial Fibrillation Related Coronary Embolism: Diagnosis in the Focus"

_jpm, 2023, doi:10.3390/jpm13050780_

Round 1

Reviewer 1 Report

Dear authors,

I have received for review your proposed article entitled Diagnostic Difficulties in Atrial Fibrillation Related Coronary Embolism: a Personalized Approach

I would like to appreciate the effort made by the authors in writing this manuscript. Atrial fibrillation (AFib) and acute myocardial infarction (AMI) are two pathologies with important vital and functional implications, frequently encountered in practice. Their association, with or without the presence of other related pathologies such as stroke, requires personalized management.

The manuscript proposed for review tries to bring into discussion possible methods to identify the causal relationship between AFib and embolic AMI. It also tries to assess the prevalence and means of diagnosis of CE both in the study group and in the available literature data.

However, I must point out a few issues that need to be corrected, both technical and in terms of content.

Regarding the editing:

1. I want to point out the fact that the manuscript does not follow the template proposed by the journal. This aspect needs to be rectified.

2. Line 8 - for text uniformity, I suggest replacing the numerals mentioned in the text with numbers - instead of "Four to five percent of MI cases" I suggest 4-5% of MI cases.

3. Row 87 - as in the previous comment, I suggest the wording "60 (5.1%)".

4. Concerning Figure 3 - it needs to be revised by improving image quality and removing the underlining.

5. Regarding Figure 2 - it needs to be revised to improve image quality.

6. Concerning Figure 3, the same observation as above. I suggest that instead of using a "PrtScr" method, the figure should be created in a program at the authors' discretion and saved as an image.

7. Regarding Figure 4, the same observation as for Figure 1. I also suggest changing the figure in a space-saving manner.

8. Line 183: at the end of the sentence there is an extra space.

9. In line 144 - an observation valid for the whole manuscript - some abbreviations are mentioned without prior explanation. Incorrectly, these abbreviations are explained later on - line 199. Please correct this.

10. In line 149, the abbreviation PAF is used improperly and redundantly, the meaning given to the phrase being "previously known previously known atrial fibrillation".

Regarding the content of the manuscript:

1. The introduction is far too broad and does not highlight the purpose of the study or the elements of novelty and originality. Furthermore, it presents literature data from a period of time that records significant diagnostic and management differences from the present time.

2. Regarding the higher frequency of cases with previously known versus newly diagnosed AFib, could you provide an explanation?

3. Regarding the observation "CHA2DS2-VASc score was significantly higher in PAF compared to NOAF patients (3.7 vs. 2.7, p<0.0001)." could you provide an explanation?

4. Regarding the comparison with the cases available in the literature, I suggest considering dropping this altogether. Although the idea may be of interest the added value for the present study is questionable.

5. Line 178: "CHA2DS2-VASc score can give the embolic risk in AF but it is able to predict new onset AF, however it is not routinely used for prediction in clinical practice." The given sentence is unintelligible. I recommend rephrasing and explaining it.

5. In the discussion section, I recommend refraining from wording such as "we believe that..." as long as the results of the study do not support them.

6. I suggest rewording the title. In its current form, it does not accurately express the content of the manuscript.

7. As a general impression, the proposed study shows scientific potential, but the given manuscript does not emphasize the novelty elements or the perspectives it could open. The choice to apply the same methods to both the study cohort and a cohort drawn from the literature will likely lead to confusion about the type of manuscript. Moreover, this choice does not bring significant elements in achieving the purpose of the manuscript.

In this context, I consider the manuscript to be acceptable for publication, but only after a major revision. I look forward to the revised version.

Kind regards

Author Response

Dear Reviewer,

I would like to say thank you for your precise and detailed review that helped to make the manuscript better. Proposed corrections in editing were changed. Regarding evaluation of CE cases published in the literature I feel must keep. These cases validate and confirm that refinement of proposed criteria of Shibata are necessary. All the other suggested changes were performed. 

Sincerely Yours,

László Balogh

Reviewer 2 Report

Balogh and colleagues presented a retrospective series of 1181 patients with STEMI in order to investigate to role of atrial fibrillation and potential coronary embolism in the genesis myocardial infarction. Therefore, the authors based their analysis on the proposed diagnosis criteria for coronary embolism published by Shibata et al. (Shibata et al. Prevalence, Clinical Features, and Prognosis of Acute Myocardial Infarction Attributable to Coronary Artery Embolism. Circulation 2015;132(4):241-50). They also took the opportunity to perform a literature review to identify coronary-embolism related myocardial infarction case reports. The article is in average well-written. The real relevance of such investigation should be to better identify the patients at high-risk fir coronary embolism and to improve the management of myocardial infarction related to coronary embolism (aspiration? Thrombolysis?). A personalized diagnosis approach should lead to a personalized management. The article is mainly descriptive. Based to their findings, what could the authors recommend to manage myocardial infarction due to atrial fibrillation related coronary embolism?

Herewith enclosed my main comments:

-       Some passage in the text miss grammatical punctuation (page 1, lines 26-28 ; page 1, lines 30-31; page 1, lines 32-33; page 6, lines 184-185). 

-       Please add the reference to the Shibata’s criteria in the article. 

-       The supplemental references is the same as the common reference list, so please, avoid it.

-       Page 6, lines 184-185. The authors refers to the current guidelines to state that aspiration is not prohibited in patients with acute coronary syndromes undergoing percutaneous coronary intervention. However, the cited references refers to a case-report (Ref 14. Zasada et al.). Did the authors mean the 2021 ACC/AHA Revascularization Guidelines? The article did not mention any guidelines at all, can you please add the most recent guidelines about thrombus aspiration? 

-       Did all patients undergo thrombus aspiration in this cohort? 

-       What was the addictive value of the diagnosis in the management of the myocardial infarction? The discussion miss some extrapolation on how the accuracy of the diagnosis could impact the way we deal with myocardial infarction in patients with AF who may suffer from coronary embolism?  

-       I would present the purpose of the article as a way to (re)-integrate thrombus aspiration in a specific population presenting myocardial infarction, more than a retrospective description of the prevalence of atrial fibrillation related coronary embolism among patients presenting with myocardial infarction. 

-       The article emphasizes a certain limitation of the Shibata’s criteria. What could you propone as other diagnosis method ?

Author Response

Dear Reviewer,

I would like to say thank you for your review that helped to make the manuscript better. Proposed corrections in editing and content have been changed. The only question that cannot be incorporated/answered in the manuscript is the proposed refined coronary embolism (CE) diagnostic criteria system. We are preparing a score system which hopefully serves better diagnostic accuracy regarding CE. We are planning to publish it later.

  1. The reference list was updated.
  2. Confirmed diagnosis of CE (in AF patient as well) interventional cardiologist must be consider thrombus aspiration to avoid unnecessary stenting and inapropriate secundary perevention drug treatment.
  3. In the whole cohort thrombus aspiration was performed 24.9%, in AF cohort 27.4% of the cases.

Sincerely Yours,

László Balogh

Reviewer 3 Report

The authors have nicely addressed the diagnostic challenges in coronary embolism related atrial fibrillation and compared their findings with the Shibata's criteria. My suggestion is to improve the presentation of the findings using graphs or other methods. In addition, discussion should provide some recommendations based on their data that may be helpful to improve the existing diagnostic criteria system.

Author Response

Dear Reviewer,

I would like to say thank you for your review that helped to make the manuscript better and more focused. The only question that cannot be incorporated/answered in the manuscript is the proposed refined coronary embolism (CE) diagnostic criteria system. We are preparing a score system which hopefully serves better diagnostic accuracy regarding CE. We are planning to publish it later.

Sincerely Yours,

László Balogh

Round 2

Reviewer 1 Report

Dear authors,

I would like to appreciate the effort made in correcting the manuscript. 

Best of luck in your future works!

Kind regards. 

Author Response

Dear Reviewer,

I would like thank you again for your review that really helped to make the manuscript better, more fluent and focused. 

Sincerely Yours,

László Balogh